# Telomere-to-telomere assemblies of cattle and sheep Y-chromosomes uncover divergent structure and gene content

Temitayo A. Olagunju[1], Benjamin D. Rosen[2], Holly L. Neibergs[3], Gabrielle M. Becker[1], Kimberly M. Davenport[3], Christine G. Elsik[4], Tracy S. Hadfield[5], Sergey Koren[6], Kristen L. Kuhn[7], Arang Rhie[6], Katie A. Shira[1], Amy L. Skibiel[1], Morgan R. Stegemiller[1], Jacob W. Thorne[8], Patricia Villamediana[9], Noelle E. Cockett[5], Brenda M. Murdoch[1] ✉ & Timothy P. L. Smith[7] ✉

Reference genomes of cattle and sheep have lacked contiguous assemblies of the sex-determining Y chromosome. Here, we assemble complete and gapless telomere to telomere (T2T) Y chromosomes for these species. We find that the pseudo-autosomal regions are similar in length, but the total chromosome size is substantially different, with the cattle Y more than twice the length of the sheep Y. The length disparity is accounted for by expanded ampliconic region in cattle. The genic amplification in cattle contrasts with pseudogenization in sheep suggesting opposite evolutionary mechanisms since their divergence 19MYA. The centromeres also differ dramatically despite the close relationship between these species at the overall genome sequence level. These Y chromosomes have been added to the current reference assemblies in GenBank opening new opportunities for the study of evolution and variation while supporting efforts to improve sustainability in these important livestock species that generally use sire-driven genetic improvement strategies.

The suppression of recombination between the mammalian X- and Y-chromosomes[1,2] outside the pseudo autosomal region (PAR) followed their separation from autosomes about 190MYA[3]. The X-chromosome gene content was maintained while the Y-chromosome rapidly lost genetic content[4,5] while accumulating duplicated DNA elements and repeats. The loss of genes on the Y-chromosome was followed by the acquisition of male-specific genes which are critical for sex determination of an individual and play vital roles in spermatogenesis and male fertility[6–10].

Obtaining a complete assembly of mammalian Y-chromosome has been elusive mainly due to the high repetitive DNA content and the inability of sequencing technologies and assembly tools to sufficiently tackle the challenges presented by the structure of this sex-determining chromosome. Very few mammals, including the *Bovidae* family to which cattle and sheep belong, have had a Y-chromosome assembly to date, and no member of the *Bovidae* family has a complete Y-chromosome. The majority of previous attempts to characterize the Y-chromosomes of the *Bovidae* family have been based on

[1]Department of Animal, Veterinary and Food Sciences (AVFS), University of Idaho, Moscow, ID, USA. [2]Animal Genomics and Improvement Laboratory (AGIL), ARS, USDA, Beltsville, MD, USA. [3]Department of Animal Sciences, Washington State University, Pullman, WA, USA. [4]Divisions of Animal Sciences and Plant Science & Technology, University of Missouri, Columbia, MO, USA. [5]Animal, Dairy and Veterinary Sciences (ADVS), Utah State University, Logan, UT, USA. [6]Genome Informatics Section, Center for Genomics and Data Science Research, National Human Genome Research Institute, National Institutes of Health, Bethesda, MD, USA. [7]U.S. Meat Animal Research Center (USMARC), ARS, USDA, Clay Center, NE, USA. [8]Texas A&M AgriLife Extension, San Angelo, TX, USA. [9]Department of Dairy and Food Science, South Dakota State University, Brookings, SD, USA. ✉e-mail: bmurdoch@uidaho.edu; tim.smith2@usda.gov

fluorescence in situ hybridization (FISH)[11]. Relatively non-contiguous 40 megabase[2] (Mb) and 43.3 Mb (NCBI accession GCA_000003205.6) assemblies of the cattle Y chromosome[2] have been produced from bacteria artificial chromosome (BAC) clones, in addition to a 16 Mb Y chromosome assembled in 67 contigs[12] from long reads sequencing, out of the estimated 50 Mb size[13]. A 10.8 Mb male-specific Y (or MSY) assembly comprised of 50 contigs alongside 4.11 Mb of the PAR has been produced for sheep[14].

The first T2T assembly of the complete human genome[15] did not have a complete Y-chromosome assembly due to the use of a cell line lacking a Y, although an assembly for this chromosome from another source was recently added[15]. The successes recorded in various T2T chromosome assemblies[16,17] have been made possible by recent advances in long read sequencing technologies[18] complemented by improvements in genome assembly algorithms[19], effectively bridging the technological gap that previously hindered successful sequencing and assembly of Y-chromosomes. Furthermore, the use of parental data in genome assembly introduced with the trio-binning method[20] has been an invaluable technique to produce fully phased haplotype-resolved assemblies of diploid species.

In this work, we present complete and gapless T2T assemblies of cattle and sheep Y-chromosomes. These sex chromosomes were obtained from haplotype-resolved whole-genome assemblies based on a combination of Illumina short reads[21] and Pacific Biosciences[22] and Oxford Nanopore Technology[23] long read sequencing technologies. We present a detailed structural analysis of these chromosomes highlighting novel features in hitherto hard to reach regions, and further elucidate the similarities and differences between them. These complete Y-chromosomes of cattle and sheep provide important resources for studying ruminant biology and mammals by extension. By interrogating these T2T Y-chromosomes we can begin to address salient long-standing biological questions around the structure and evolution of the Y-chromosomes of these two members of the *Bovidae* family.

## Results

### Whole genome assemblies of cattle and sheep
The T2T Y-chromosome assemblies of cattle and sheep were obtained from draft versions of haplotype-phased whole genome draft assemblies (in progress) of the 120-day and 100-day gestation F1 individuals from the Sire_x_Dam crosses of Wagyu_x_Charolais cattle and Churro_x_Friesian sheep breeds respectively, using the Verkko[19] assembler and parental data for phasing. The cattle Y-chromosome thus represents a Wagyu haplotype while the sheep Y-chromosome is from a Churro. The combination of the ONT ultra-long (UL) reads with the PacBio Hifi reads successfully resolved the highly repetitive telomeres, centromeric and heterochromatic regions of the Y-chromosomes. For cattle, about 626.4 Gbp raw ONT reads with 129.4 Gbp of UL reads at 101× and 21× respective reads depth were used for the genome assembly, while 328.6 Gbp of PacBio HiFi raw reads at 53× were used (Supplementary Data S1). Similarly for sheep, 600.8 Gbp of raw ONT reads at 106x with 15x of 84.6 Gbp UL reads were combined with 258.1 Gbp at 46× for the assembly (Supplementary Data S1). The Y-containing paternal haplotype assemblies were highly contiguous with contig N50 of 96.68 Mb (cattle) and 108.17 Mb (sheep) (Supplementary Data S2). The reference-agnostic genome assembly completeness evaluation with Merqury[24] revealed high rate of k-mer survival of the raw reads in the final assemblies (Supplementary Fig. S1) with 53.6–56.13 QV scores, while 98.11–99.78% complete BUSCOs were discovered from the *Cetatiodactyla_ODB10* orthologous genes database (Supplementary Data S2). Visualization of the haplotype-colored assembly graphs with Bandage[25] showed that some of the autosomes were not gapless, but the cattle and the sheep Y-chromosomes were in single contigs (Supplementary Fig. S2). The Y chromosome contigs were extracted from the paternal haplotype assemblies for in depth

analysis. Merqury[24] QV scores for the cattle and sheep Y-chromosome contigs were higher than their paternal haplotype averages at 62.38 and 59.95, respectively (Supplementary Data S2). Telomere sequences were located at the distal ends of the two chromosomes—14.7 kb and 20.3 kb from the p- and q- arms of the cattle and 19.3 kb and 17.8 kb from the p- and q- arms of the sheep, (Table 1, Fig. 1A) indicating the completeness of the single-contig assemblies. The telomeric regions were characterized by a tandem repeat of the 6-mer sequence CCCTAA on the p-arm and the reverse complement TTAGGG on the q-arm of both Y-chromosomes. The cattle Y-chromosome contained 2447 copies of the complete sequence on the p-arm and 3387 copies on the q-arm. Similarly, the sheep Y-chromosome harbored 3211 copies of the 6-mer on the p-arm while the q-arm contained 2962 copies.

### The cattle and sheep Y-chromosomes have substantial differences in structure
The total lengths of the complete cattle and sheep T2T Y-chromosome assemblies were substantially different at 59.4 Mb and 25.9 Mb, respectively. The 120.76 kb sheep Y-chromosome centromere at 8.03 Mb from the end of the short arm is 20 times smaller than the 2.52Mb-long centromere on the cattle located at 14.12 Mb from the end of the shorter arm of the chromosome (Table 1, Fig. 1A). Approximately half (50.54% for cattle and 48.55% for sheep) of the bases on the two chromosomes were annotated as repetitive DNA (Fig. 1B, Supplementary Data S3), and the q-arms comprised a long region of a mosaic of high similarity repetitive DNA arrayed in either direct or inverted orientation to one another (Fig. 1A). The distribution of the repeat elements indicates that LINE elements had the highest and similar proportion (about 31%) on both Y-chromosomes (Fig. 1B, Supplementary Data S3). The proportion of other classes of repetitive DNA were higher in sheep except for satellites and simple repeats (low complexity region) (Fig. 1B, Supplementary Data S3). This observation agrees with previous reports that LINE elements are the dominant retrotransposons in mammals[26,27].

The PAR on the Y-chromosome assemblies were identified by mapping long reads from female haplotypes described in ref. 28 with further details in the methods section. The alignment track visualized with IGV[29] clearly defined the PAR boundaries with soft clipping of the long reads at the region lacking homology with the Y-chromosome, rendered as the highly colored reads segment coupled with a drastic drop of the coverage to zero on the coverage track (Supplementary Fig. S3). The sheep Y-chromosome PAR (7,018,329 bp) was 195.9 kb longer than the cattle PAR (6,822,380 bp) (Fig. 1A, Table 1) despite having much lower total chromosome length. The rest of the Y-chromosome adjacent to and outside the PAR is the MSY region comprising the gene-rich euchromatin and gene-deficient heterochromatin of the chromosome (Fig. 1A). This region is sub-classified into the X-degenerate region harboring the genes that ancestrally recombined with the X-chromosome, and the ampliconic region containing intrachromosomal duplication of genes which are expressed mainly in the testis[30].

The Y-chromosome assemblies were submitted to NCBI and annotated with their Eukaryotic Genome Annotation Pipeline (https://www.ncbi.nlm.nih.gov/genome/annotation_euk/process/) for use in downstream analyses. The number of protein-coding genes on the cattle Y-chromosome was more than threefold higher than that found on the sheep Y-chromosome (352 to 109) (Supplementary Data S5 and S6). However, more pseudogenes (150 to 79) were annotated on the sheep Y compared to the cattle. This difference in the number of annotated genes, with an approximate 1:2 ratio is proportionate to the chromosome sizes. All the previously reported mammalian PAR genes identified through sequence analysis[31] and physical mapping with FISH[11] could be identified on each T2T assembly. The most proximal protein-coding gene in the PAR of both Y chromosomes was *PLCXD1*, located at 33.74 kb on the sheep and 34.57 kb on the cattle from the telomeres on

**Table 1 | The structure and gene content of the cattle and sheep Y-chromosomes**

| Cattle | | | | | Sheep | | |
|---|---|---|---|---|---|---|---|
| **Chromosome structure** | | | | | | | |
| | Feature | length (bp) | start | end | length (bp) | start | end |
| | Telomere-p | 14,685 | 1 | 14,685 | 19,267 | 1 | 19,267 |
| | PAR | 6,807,694 | 14,686 | 6,822,380 | 6,999,061 | 19,268 | 7,018,329 |
| | MSY | 52,633,132 | 6,822,831 | 59,455,963 | 18,880,972 | 7,018,330 | 25,899,302 |
| | Telomere-q | 20,325 | 59,455,964 | 59,476,289 | 17,775 | 25,899,303 | 25,917,078 |
| | centromere | 2,521,056 | 14,124,633 | 16,645,689 | 120,763 | 8,038,393 | 8,159,156 |
| **Annotation** | | | | | | | |
| | Protein-coding (PAR, X-d, Amp) | 236 (34, 15, 187) | | | 109 (45, 18, 46) | | |
| | Pseudogenes (PAR, X-d, Amp) | 50 (2, 20, 28) | | | 127 (0, 23, 104) | | |
| | lncRNA | 14 | | | 29 | | |
| | tRNA | 9 | | | 7 | | |
| | snRNA | 5 | | | 5 | | |
| | snoRNA | 1 | | | 1 | | |
| **Multi-copy genes** | | | | | | | |
| PAR genes | *ASMTL* | 2 | | | 2 | | |
| | *OBP* | 3 | | | 4 | | |
| | *BOS2D* | 2 | | | 4 | | |
| | *CSF2RA* | 1 | | | 2 | | |
| X-d genes | *USP9X* | 1 | | | 2 | | |
| Ampliconic genes | *HSFY (Protein-coding, pseudogene)* | 40 (37, 3) | | | 26 (12, 14) | | |
| | *HSFY2* | 2 (2, 0) | | | 0 (0, 0) | | |
| | *PRAME* | 31 (31, 0) | | | 6 (1, 5) | | |
| | *RBMY* | 11 (11, 0) | | | 1 (1, 0) | | |
| | *TSPY1* | 82 (68, 14) | | | 52 (0, 52) | | |
| | *TSPY3* | 16 (16, 0) | | | 24 (5, 19) | | |
| | *ZNF280B* | 40 (22, 18) | | | 42 (27, 15) | | |
| | *ZNF280A* | 15 (0, 15) | | | 22 (0, 22) | | |

X-d = X-degenerate.

the p-arms of the respective Y-chromosomes while the most distal gene was *GPR143* located about 3.50 kb from the PAR boundaries of both Y-chromosomes (Supplementary Data S5 and S6). The PAR on the sheep harbored a total of 45 protein-coding genes comprising 33 single-copy genes and four multi-copy genes (*CSF2RA*:2, *ASMTL*:2, *OBP*:4 and *BOS2D*:4) (Supplementary Data S5) while 27 single-copy and three multi-copy genes (*ASMTL*:2, *OBP*:3 and *BOS2D*:2), a total of 34 protein-coding genes, were located on the cattle PAR (Supplementary Data S4). These included an uncharacterized gene *LOC112445918* (Supplementary Data S4 and S6) in the cattle Y PAR, and copies of Splicing Factor 3 A Subunit 2 (*SF3A2*) and Proline Rich Protein BstNI Subfamily 1 and 3 (*PRB1* and *PRB3*) (Supplementary Data S5) on the sheep Y PAR which were not in the PAR of their respective X-chromosomes (Supplementary Data S6). The overall conservation of the mammalian PAR genes on both cattle and sheep Y-chromosomes is expected since crossing over still occurs between the X- and Y-chromosomes.

The MSY contains the sex-determining gene *SRY* and is further divided into sub-regions named the X-degenerate (X-d) and ampliconic regions. The sex-determining gene, *SRY* is located at 58,122,906 bp and 17,199,770 bp on the cattle and sheep Y-chromosomes, respectively (Supplementary Data S4 and S5). The X-d region contains genes that still maintain some level of homology with the X-chromosome outside the PAR. The cattle X-d (spanning about 1.9 Mb at 6.8Mbp and 23.0Mbp from the p-arm) contained 15 protein-coding genes with 3 having pseudogenes (*EIF1AX*:1, *UBA*:2, *SHROOM2*:3) for a total of 27 pseudo-genes (Supplementary Data S4). Out of the 18 protein-coding genes in the sheep X-d region (about 1.7 Mb long at 7.0Mbp from the p-arm),

only *USP9X* is multi-copy with just 2 copies; *EIF1AX* is the only protein-coding gene having pseudogenes with 4 copies out of the total 23 pseudogenes in this region (Supplementary Data S5).

The putative ampliconic gene families were extracted from the NCBI annotations including *RBMY*, *HSFY*, and *TSPY* that are conserved across mammals[32] as well as the bovine specific genes *PRAME*, *ZNF280A* and *ZNF280B*[7,33]. There were approximately fourfold more cattle protein-coding ampliconic genes (187) compared to the sheep (46) (Fig. 2A, Supplementary Data S7). However, like the overall observation on the gene content, the pseudogenes on the sheep (127) were more than double the number on the cattle (50) (Supplementary Data S7). These figures highlight significant divergence in the copy number of the ampliconic genes between cattle and sheep. Bearing in mind that the size and content of the PAR of the two chromosomes are similar (Fig. 1A, Table 1), and the number of X-degenerate genes are also similar (Supplementary Data S4 and S5), the ampliconic genes, occupying an approximate 40MB block of DNA on the cattle compared to about 17MB on the sheep, appear to be responsible for the large chromosome size difference.

Ampliconic genes are important MSY genes which are required for spermatogenesis and fertility[34–36]. Variation in their copy number within a population has been previously reported in cattle for *TSPY*[37,38], *PRAME*[39] and *ZNF280A*[40,41], and copy number has been implicated in male fertility[39,42,43]. Ampliconic genes were present mainly in tandem arrays on the cattle, but not on the sheep. For instance, *HSFY*, *RBMY*, *PRAME* and *TSPY* were arranged in tandem on the same strand on the cattle, whereas only 1 copy each of *RBMY* and *PRAME* were on the sheep

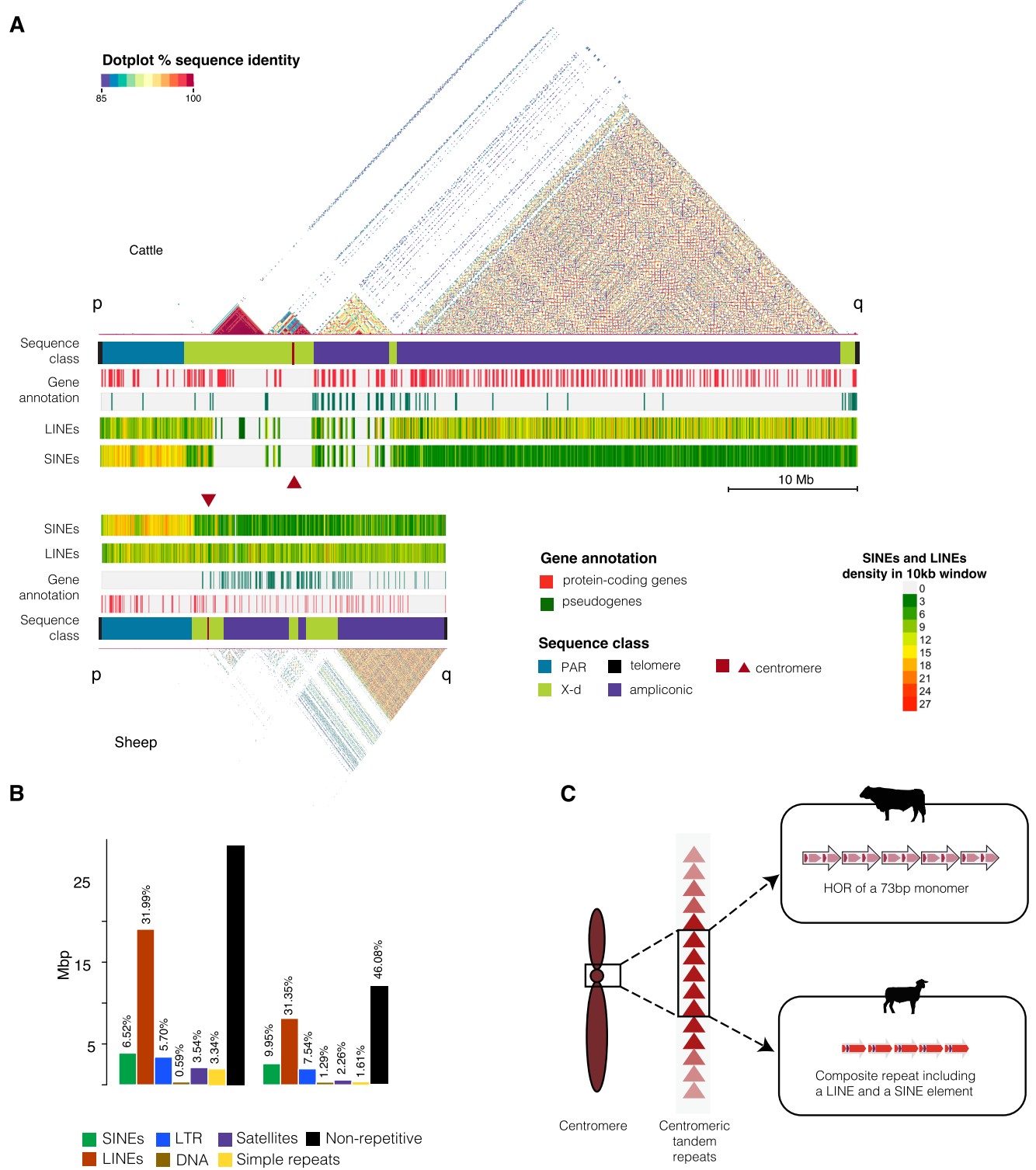

**Fig. 1 | Global structure of the cattle and sheep Y chromosomes.** The structure of the cattle and the sheep Y-chromosomes showing the differences and the similarities between the two species. **A** The self-identity dotplots highlight the mosaic of repetitive sequence on the q-arms while the tracks below it show the different sequence classes, gene annotation, LINEs and SINEs. **B** The repeat content annotation indicates that more than half of both Y-chromosomes is repetitive DNA.

**C** Centromere content and organization shows that the two Y-chromosomes have tandemly arrayed repeat unit at the centromere; while the cattle centromeric repeat is organized into a higher order repeat (HOR) of a 73 bp monomer, the sheep repeat unit is a composite of a tandem LINE and SINE element with spacer DNA between the copies. Source data are provided as a Source Data file.

(Fig. 2B and Supplementary Fig. S4). An island of 44 protein-coding copies (out of the total 84 copies) of *TSPY* is located on the p-arm of the cattle spanning 1.21 Mb while the rest of the copies are distributed on the ampliconic region of the chromosome (Supplementary Fig. S4A).

*TSPY* has been reported as the largest tandem protein-coding array present on the human genome[15,32,44] and the active array constitute the active copies, but they have been more amplified on cattle relative to human[41].

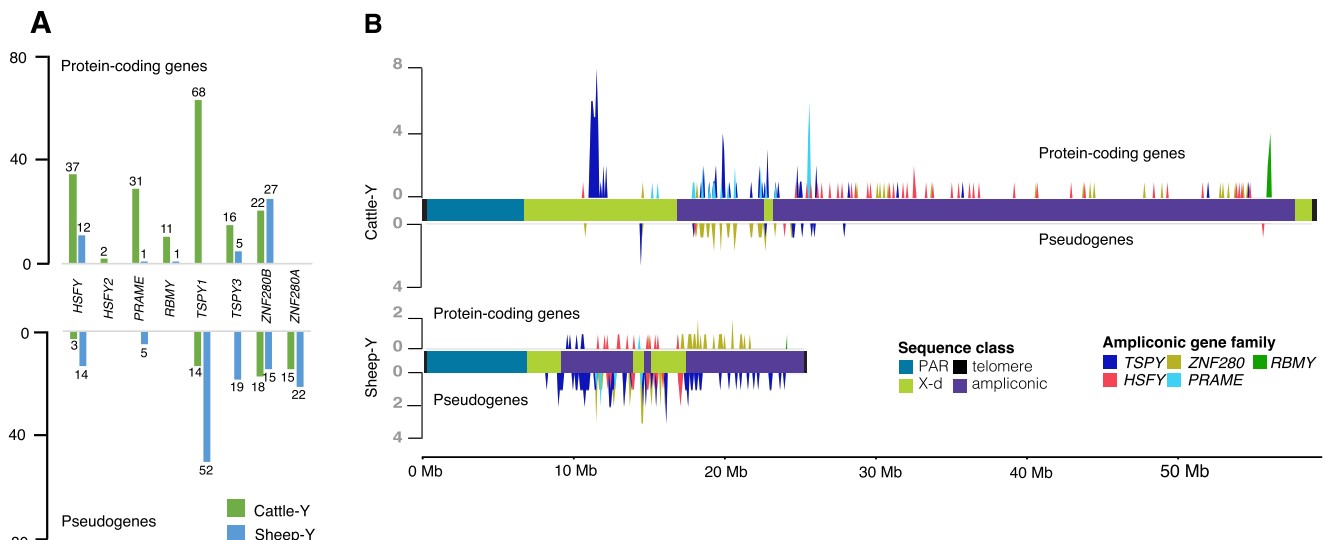

**Fig. 2 | Ampliconic genes content comparison between cattle and sheep. A** The copy distribution of the protein-coding and the pseudogene copies of the ampliconic genes families on the cattle and sheep Y-chromosomes show remarkable differences (Source data are provided as a Source Data file). **B** The gene density showing the number of ampliconic genes across the Y-chromosomes calculated in 100 kb bin sizes highlighting the loci of the protein-coding genes on the upper tracks and the pseudogenes on the lower tracks. There are no protein-coding copies of TSPY1 on the Sheep-Y on the lower panel a TSPY1 island (high blue clustered spike) is located on the cattle-Y harboring a tandem array of 44 copies out of the total 68 protein-coding copies on the chromosome. More pseudogenes are found on the Sheep-Y relative to the protein-coding genes suggesting that some of the ampliconic genes are being pseudogenized, losing their protein-coding capabilities.

## Pseudogenization of ampliconic genes suggests an evolutionary process on the Y-chromosomes

A lower ratio of protein-coding genes to pseudogenes in the ampliconic gene family was observed on the sheep (0.07 to 0.85) compared to the cattle Y-chromosome (0.66 to 31) (Fig. 2A, Supplementary Data S7). The *TSPY* gene family had the lowest ratio (0.07) in sheep. Only *ZNF280B* ampliconic gene had more protein-coding copies (1.80 ratio) than pseudogenes (Fig. 2B, Supplementary Data S7). With no protein-coding copy of *ZNF280A* annotated on either Y-chromosome, 15 and 22 copies of the pseudogene were present on the cattle and the sheep respectively. In contrast, no *RBMY* pseudogenes were found on either chromosome (Fig. 2B, Supplementary Data S7 and Supplementary Fig. S4).

A previous study on the evolution of the bovine MSY genes, based on the expansion of the ampliconic genes[41], reported that the amplification of the *TSPY* and *ZNF280B* genes predated the amplification of *HSFY* genes around the same time that sheep diverged from cattle[45]. Since the amplification of *TSPY* occurred before the divergence of the two species, it is therefore expected that a relatively high copy number similar to the number found on the cattle (98) should also be found on the sheep. However, the preponderance of pseudogenes (71) to protein-coding copies (5) (Fig. 2A, B, Supplementary Data S7) suggests the decay of their protein-coding copies on the sheep Y in line with chromosome evolution theory of gene loss through pseudogenization[32]. This may also explain the observed higher pseudogene copies of *HSFY* and *PRAME* than the protein-coding ones (Fig. 2A, B, Supplementary Data S7). In contrast to the other ampliconic gene family members, *ZNF280B* still maintained more protein-coding copies than pseudogenes (Fig. 2A, B, Supplementary Data S7).

A cross-species phylogenetic comparison of the protein-coding copies of the ampliconic gene families revealed a general high intra-species clustering between the genes with clearly separated clades for the cattle and the sheep copies (Supplementary Fig. S5). The protein-coding copies of *ZNF280B* did not show a clear separation from the ancestral autosomal copy[33] (Supplementary Fig. S5A), while the higher copies on sheep may have resulted from interlocus gene conversion events. In contrast, the *HSFY* and *TSPY* copies on cattle and sheep branched at the human copy-rooted trees (Supplementary Figs. S5B & S5C). The fewer protein-coding copies on sheep compared to cattle is suggestive of loss of the ancestral copies on sheep in response to evolutionary pressures since sheep diverged from cattle[45]. *RBMY* and *PRAME* were not included in this cross-species phylogenetic comparison since the sheep Y had just one copy of each.

The transcriptional activity of the ampliconic gene families was investigated with transcript reads from RNA-Seq experiments of different tissues and developmental stages (Supplementary Data S8 and S9) and indicated that most of the protein-coding copies of these genes had no evidence of expression. On the cattle Y-chromosome, *TSPY* with the highest protein-coding copies (84) had only 7 copies with observed transcripts while a single copy of *TSPY3* showed considerable levels of transcriptional activity. Only 1 out of the 11 copies in the *RBMY* array had evidence for transcriptional activity (Supplementary Data S8). Similarly, 18 protein-coding copies of the total 31 for *PRAME* recorded reads while only 9 copies were at substantial levels. *ZNF280B* was the only gene with transcriptional activity across all the tissues analyzed with transcripts corresponding to 14 out of 23 gene copies and only 3 of these displaying relatively high transcriptional activity (Supplementary Data S8). It is noteworthy that the ancestral copies of the ampliconic genes showed less transcriptional activity compared to the bovine specific *PRAME* and *ZNF280B*. The significance of this is not understood yet but will require further studies.

All the copies of the protein-coding ampliconic genes on the sheep generally exhibited high levels of transcriptional activity except 2 of the 27 copies of *ZNF280B* which registered no reads across all the samples analyzed (Supplementary Data S9). These results indicate that only 10% of the protein-coding copies of the ampliconic genes on the cattle registered transcriptional activity in sharp contrast to 91% on the sheep. Low gene expression levels of ampliconic gene families have been linked to their nonessentiality on the Y-chromosome of apes[46], evolution of new function[47] or compensation by the X-chromosome paralog[47]. While there is no sufficient data in this study to draw any of these conlusions on the observed differences in the transcriptional activities of the ampliconic genes, the preliminary observation is suggestive of some ongoing evolutionary mechanism on the Y chromosomes.

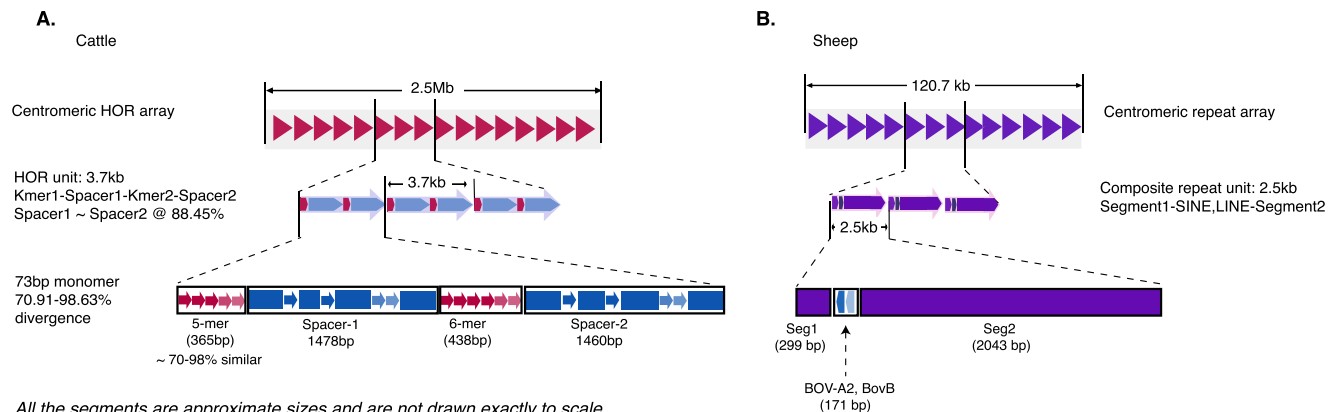

**Fig. 3 | Content and organization of the Y-chromosome centromeres. A** The Cattle-Y centromere is organized as a tandem array of highly identical higher-order repeat (HOR) unit spanning 2.5 Mb. The 3.7 kb HOR contains copies of a 73 bp monomeric unit arranged into four segments where segments 1 and 3 are tandem arrays of 5 copies (5-mer) and 6 copies (6-mer) respectively in red color while segments 2 and 4 in light blue color contain copies of the monomeric unit but not in tandem arrays as in segments 1 and 3. The arrows representing the monomers have different color densities to highlight the divergence between them. **B** The Sheep-Y centromere spanning about 120.7 kb organized as an array of a 2.5 kb composite repeat unit comprising a BOV-A2 SINE and a BovB LINE tranposable elements embedded between two segments of DNA.

## Novel insights into the cattle and sheep centromere structure and organization

The physical coupling of chromosomes to spindle fibers during cell division is facilitated by assembly of the kinetochore proteins at the centromere. The centromeres of most eukaryotic chromosomes are characterized by satellite DNA repeats which are generally involved in a higher order repeat (HOR) structure of composite elements[48] as the genetic signal for kinetochore attachment. A characteristic epigenetic methylation dip at the centromeric dip region (CDR) due to the binding of the centromere protein A (CENP-A) has also been observed[49]. A generally high CpG methylation within the active centromeric alpha satellite array relative to the adjacent regions is interrupted by local reduced CpG methylation dips at the CDRs[50].

The cattle centromere was defined by a well-organized tandem array of a HOR containing a 73 bp monomeric satellite repeat unit (Fig. 1C). This is the first time that this sequence has been identified in cattle. The 73 bp monomeric satellite unit was arranged into a HOR structure wherein a total of 364 copies of the 3.7 kb unit spans the 2.5 Mb centromeric region (Fig. 3A). The individual HORs had 95–98% sequence identity between them and were organized into four segments based on the homogenization of the tandem monomeric repeat unit – the first and third segments comprised 71-98% identical 5-mer and 6-mer tandem copies of the 73 bp monomers, respectively, while the second and fourth segments contained more diverged copies of the monomer and non-satellite DNA (Fig. 3A). There were other copies of the HOR found on the cattle outside the centromere, but they were more diverged from the copies within it (Supplementary Fig. S6). Similarly, more divergent copies of the 73 bp monomeric unit were evenly distributed along the entire length of the cattle MSY region (Supplementary Fig. S7A). Using cytosine residue methylation status predicted from the long sequence read data produced by the ONT PromethION and PacBio Revio platforms, the CDRs supporting the annotation of the centromere locus on the cattle were revealed at the flanks of the centromeric HOR array and coincided with an enrichment of signal from the inner kinetochore CENP-A (Supplementary Fig. S8).

The alignment of the cattle 73 bp centromeric monomer to the whole genome assemblies (comprising the two haplotypes each) of the Churro and the Wagyu suggests that the sequence is specific to only the sex chromosomes but with much fewer and more diverged copies on the cattle and sheep X-chromosomes and the sheep Y-chromosome (Supplementary Fig. S7B, Supplementary Data S10).

The sheep Y-chromosome centromere was organized differently than the cattle as a tandem array of 47 units of a 2.51 kb composite structure comprising two ruminant-specific transposable elements (TEs) *BOV-A2* and *BovB* interspersed with spacer sequences (Fig. 1C). The composite structure can be subdivided into three segments – a 299 bp segment1, 171 bp segment2 and 2043 bp segment3 (Fig. 3B). Segment2 contained RepeatMasker-annotated *BOV-A2* SINE and *BovB* LINE TEs. This tandem co-location of *BOV-A2* and *BovB* is not surprising since LINEs have been reported to facilitate the transposition of SINE elements[51]. A C-rich 58 bp simple repeat sequence at the end of segment2 was the only other annotated repeat, segments 1 and 3 were lacking any annotation. The methylated cytosine data supported identification of the characteristic CDR which defines the site of kinetochore assembly on eukaryotic centromeres as epigenetic support for the annotation of the sheep Y-chromosome centromere (Supplementary Fig. S9).

The presence of TEs within the repeat unit led us to hypothesize that the TE may have been inserted into a piece of DNA before replication and homogenization at the centromere. This hypothesis was tested by mapping the sequence surrounding the TEs to the whole genome to locate a possible origin of the repeat unit. However, there was no significant mapping and as such the origin of the repeat unit could not be ascertained in this manner.

Most of the putative bovine satellite DNA sequences have been reported to be related to one another with short segments[52] arising from their evolutionary trajectory. For instance, *SAT1.706*, *SAT1.711a* and *SAT1.720* share a 23 bp sub-repeat unit while *SAT1.711b* and *SAT1.715* also share another 31 bp sub-repeat unit. Based on this, we sought to identify any relatedness of the cattle and sheep centromeric repeat sequences (Supplementary Data S11) to any of the putative bovine satellites by aligning them, but none was observed. This suggests that they are novel sequences and bear no evolutionary relationship to the known bovine satellites.

## The T2T Y-chromosomes compared with publicly available cattle and sheep assemblies

Our T2T Y-chromosome assemblies were compared with publicly available assemblies of cattle and sheep on NCBI to assess the concordance of the available assemblies with our T2T assemblies. A 43.3 Mb assembly of Hereford cattle Y chromosome (BTAU5-Y) with the accession GCA_000003205.6 and a Hu sheep assembly with

accession CM022046.1 from a previous study[14] were obtained from NCBI.

The first 2.5 Mb of the BTAU5-Y was absent and only about 100 kb sequence overlapped between the two Y-chromosome assemblies at the PAR (Supplementary Fig. S10A, Supplementary Data S12). The next alignment segment was 0.5 Mb long and was followed by a large gap spanning 7.8 Mb on the BTAU5-Y up to 16.6 Mb distal to the centromere on the T2T cattle, implying an absence of the centromere on the BTAU5-Y. The lack of centromere on the BTAU5-Y was confirmed with the absence of the novel cattle centromeric HOR (with just 4 copies) despite the abundance of the 73 bp monomer (Supplementary Fig. S10B) across the length of the chromosome. The region spanned by this BTAU5-Y gap on the T2T cattle contains the *TSPY1* array (Supplementary Data S13), a repeats-enriched 4.1 Mb segment and is adjacent to the centromeric repeat (Fig. 1). The next 6 Mb of DNA on the T2T cattle aligned with only about 400 kb of the BTAU5-Y indicating more than 5 Mb of sequence harboring 33 ampliconic genes (Supplementary Data S13) is missing relative to the T2T cattle Y-chromosome (Supplementary Fig. S10A, Supplementary Data S12). Alignment contiguity between the two Y-chromosomes reduced from 23.8 Mb to the end of the T2T cattle within the highly repetitive ampliconic region. This comparison has accounted for more than 15 Mb of sequence missing in the BTAU5-Y relative to the T2T cattle Y-chromosome.

The alignment between the Hu Sheep and the T2T sheep showed a broad sequence concordance up to about 10 Mb out of the 10.6 Mb of the Hu sheep MSY extending from the end of the T2T sheep PAR to 19.67 Mb (Supplementary Fig. S11, Supplementary Data S14). However there was a 1.85 Mb region on the T2T sheep that exhibited lower alignment contiguity with an average segment length of about 150 kb as well as a few sequence inversions. The rest of the Hu sheep MSY, about 600 kb long, mapped to the T2T sheep in segments of about 50 kb extending from 17.6 Mb within the ampliconic region to its end (Supplementary Fig. S11, Supplementary Data S14). These comparisons have revealed remarkable gaps in the public assemblies, mainly in the highly repetitive regions of the chromosomes, which have been filled by the T2T Y-chromosome assemblies.

## Discussion

The combination of long reads for contig creation and short reads for haplotype phasing has produced single contiguous high-quality Y-chromosome assemblies of two members of the *Bovidae* family – cattle and sheep – spanning the highly repetitive DNA content. Following detailed analysis, we present the structure and organization of the important genomic structures contained in these two chromosomes.

The first striking observation comparing the two chromosomes is the substantially smaller size of the sheep Y-chromosome compared to the cattle Y-chromosome. The 59.4 Mb cattle Y-chromosome was more than double the 25.9 Mb length of the sheep. Although previous reports from chromosome banding techniques of a smaller sheep Y chromosome compared to cattle[53] and a significant variation of Y chromosome within the *Bovini* tribe[54] have been made, our observation presents a clearer picture of this size difference between cattle and sheep. Greater variation in the size of the Y-chromosome relative to similar X-chromosome size across primates has recently been reported[55]. This suggests a higher susceptibility of the Y-chromosome to changes in response to evolutionary pressures than the X-chromosome. The extent of this Y chromosome polymorphism within the *Bovidae* family would be better elucidated with the availability of more complete Y-chromosome assemblies. The relatively shorter sheep Y is still nearly twice as long as the 14.9 Mb length of the longest previously reported Y-chromosome sequence of the domestic sheep[14]. Similarly, the T2T cattle Y-chromosome is 9.4 Mb more than the previously reported 50 Mb estimate[12].

Gene annotation of the T2T Y chromosomes indicates that the PAR region, which still maintains recombination with the X-chromosome, is generally well conserved between cattle and sheep in agreement with the gene content of the mammalian PAR. The uncharacterized *LOC112445918* gene located on the cattle Y-chromosome is in a region missing from the X-chromosome (Supplementary Data S6) assembly and may have been present if the assembly was complete. On the sheep Y-chromosome however, the three previously unreported genes (*LOC112445918*, *PRB1* and *PRB3*) did not have homologs on the X-chromosome PAR, nor on the autosomes.

Since the number of bases covered by the PAR on the two chromosomes are similar, and the extent of the X-degenerate regions is similar in both chromosomes, the ampliconic gene families appear to be responsible for the substantial size difference between the cattle and sheep Y chromosomes. Previous studies of the ampliconic region have reported increased tendency of drastic differences in gene content and copy number between closely related species[34,56] and even within a population[37–41,55]. It has previously been suggested that ampliconic genes were involved in the diversification of *Bovidae*[41], but the extent of their contribution to the wide divergence between the cattle and sheep Y-chromosome sizes is remarkable. Studies within a clade, such as being carried out in ruminants by the Ruminant T2T (RT2T) Consortium[57], will help to elucidate Y-chromosome polymorphism between the species in that clade.

The amplification patterns of the protein-coding ampliconic genes observed on the cattle and the pseudogenization of some copies on the sheep suggests different evolutionary mechanisms taking place within the two Y-chromosomes since the divergence of sheep from cattle about 19MYA[45]. Higher copy number for genes which are associated with important traits such as fertility is correlated with higher gene expression[58], however, the transcriptional activity analysis indicated that most (90%) of the ampliconic genes had no evidence of transcriptional activity. With higher copy numbers and reduced expression in a region devoid of crossing over with the X-chromosome, the amplification might be due to gene conversion as a means to conserve gene function[32,59]. These observations are preliminary since this study was not designed to answer these questions and will require further population-level studies across breeds and species to establish the evolutionary mechanism shaping the Y-chromosomes of cattle and sheep.

Analysis of our T2T Y-chromosome assemblies of cattle and sheep have provided novel insights into the content and organization of the centromeric region of the chromosomes. Bovine satellite DNAs have been well characterized with FISH especially in the centromeric and pericentromeric regions on the autosomes of cattle and sheep, but none of these satellites has been located on either of the sex chromosomes (reviewed in ref. 52). For the first time the monomeric (73 bp) satellite as well as the HOR sequence (3.7 kb) which characterizes the centromeres of cattle as well as the composite repeat unit at the centromeres of sheep have been identified. The copies of the cattle 73 bp cattle monomer located on the sheep Y indicate that this monomer predates the divergence of cattle and sheep. The few copies on the sheep Y which were not proximal to the centromere were highly diverged in sequence compared to the copies on the cattle Y. It is yet to be determined whether a copy of this monomer was amplified and adopted as the Y centromere in cattle or lost and degraded in sheep. This new knowledge of the structure and organization of the centromeres on the cattle and sheep Y-chromosomes is invaluable to the study of chromosome biology and evolution since centromeric satellites are regarded as the fastest evolving DNA elements[60,61].

The preliminary knowledge on cattle and sheep Y-chromosomes which has been revealed by these T2T assemblies have provided the foundation for further exploration of different aspects of Y-chromosome biology at population scale. In a recent study, the copy number of the *TSPY* array was reported to vary between human, which had 44 tandem copies, and non-human primates, with an average copy number of 18[55]. It is yet to be determined what kind of evolutionary

relationship exists between the human orthologs and the tandem copies found also on the T2T cattle Y-chromosome. Furthermore, studying the gene content on other species in the *Bovidae* family or the ruminantia sub-order[57] would enrich our knowledge of the evolution of the sex-determining chromosomes within and between lineages.

The first complete T2T Y-chromosome assemblies of cattle and sheep from the *Bovidae* family have provided a holistic insight into the structure and organization of these important sex-determining chromosomes. Comparison of the two assemblies revealed interchromosomal similarities and differences in their genetic components. Remarkable differences were noticed in the size and organization of the centromeres and the overall chromosome length. The difference in the chromosome lengths, specifically in the MSY, could be ascribed to loss of copies of ampliconic genes on the sheep Y since the cattle Y is the ancestral copy, and is suggestive of different evolutionary processes on the two chromosomes. Our hypotheses can be tested when the full suite of ruminant Y-chromosomes is available from the RT2T consortium[57]. These new assemblies which are important resources for ruminant chromosome biology have been added to the current reference assemblies of cattle and sheep on NCBI and present new opportunities to answer pertinent biological questions on the sex-determining Y-chromosome.

## Methods

### Ethics statement

This study was carried out in accordance with the University of Idaho Institutional Animal Care and Use Committee (IACUC) approved protocol with number IACUC-2020-58 for sheep and IACUC-2021-21 for cattle. The male Angus calf which produced the bovine satellite cells used for the CENP-A pull down assay was harvested under the IACUC #017090 also at the University of Idaho.

### Genome assembly quality control

GfaStats[62] was used to produce the assembly statistics used to evaluate the contiguity of the whole genome assemblies using default parameters. The reference-free k-mer-based genome assembly completeness evaluation tool Merqury[24] was run as recommended by the developers on the diploid assemblies from the F1 crosses of the cattle and sheep species to produce q-value statistics and plots to check the k-mer distribution between the raw and the final assemblies for assembly completeness assessment. Compleasm[63] (formerly miniBUSCO) was also used to evaluate assembly completeness using the number of orthologous genes identified in the draft assemblies. Compleasm was run with default parameters and lineage = "cetartiodactyla_odb10".

*Seqtk telo* module from Seqtkv1.4 (https://github.com/lh3/seqtk) was used to identify the telomere sequence at the ends of the Y-chromosome assemblies. Although the program identified the telomeric repeat sequence at the ends of the Y-chromosomes, it is difficult to ascertain if the full length of the telomeres were assembled.

### Defining the PAR boundaries

We defined the PAR on the Y-chromosome assemblies by mapping raw PacBio long reads (HiFi for cattle and CCS for sheep) from female haplotypes to the cattle and the sheep Y-chromosome assemblies as previously described[28]. Briefly, the PAR on the X-chromosomes of the cattle and sheep reference assemblies were hard-masked using the current annotation coordinates. Long reads from a female individual were mapped to the assemblies using minimap2[64] (Supplementary Methods 1). The alignment file was sorted and indexed with Samtools[65] before visualization with IGV[29]. The alignment track (Supplementary Fig. S3) clearly showed the PAR boundaries with the soft clipping of the long reads at the region lacking homology with the Y-chromosome rendered as the highly colored reads segment, coupled with a drastic drop of the reads coverage to zero as shown in the coverage track (Supplementary Fig. S3).

To further finely define the PAR region for the cattle and sheep sex chromosomes, we aligned the Y- and X-chromosome assemblies to each other with Mashmap[66] using the parameters segment length *-s 10000* and minimum identity *--pi 95*. For cattle, the PAR region on the Y-chromosome extended from the beginning of the chromosomes to 6,819,999 bp on the Y-chromosome 6,810,542 Mb on the X-chromosome with 99.38% sequence identity between them. The sequence identity of the next alignment block between the two sex chromosomes spanning 9999 bp dropped drastically to 92.79%. The sheep PAR region spanned 7,019,999 bp in two blocks from the start of the Y-chromosome and between 179 bp and 6,987,958 on the X-chromosome, at an average of 98.85% sequence identity between them, while the next alignment block of 9999 bp dropped to 95.07 sequence identity.

### X-d regions

The X-degenerate region of the Y-chromosomes comprise homologous regions with the X-chromosome outside the PAR and without active recombination with it. This region was defined by aligning the Y-chromosome to the X-chromosome with Mashmap. Mashmap was run with segment length "*-s 5000*" and percentage identity "*--pi 65*".

### Gene annotation

The protein-coding genes on the Y-chromosomes were initially annotated manually in two ways - using Liftoff v1.6.3[67] and by homology search with protein sequences from cattle and sheep, as well as from the well-annotated human and mouse genomes. Homology-based annotation was necessary to supplement the lack of Y chromosome-specific genes from the liftoff of the current cattle and sheep reference assemblies since the reference had no annotated Y. Refined annotation became available upon submission to NCBI through the application of their RefSeq annotation pipeline, revealing only slight disparities in the number and loci of genes from our manual annotations (Supplementary Data S15 and S16). However, we used the NCBI annotations for the downstream analysis.

Liftoff was run with the parameters -flank 0.0 **-s** 0.4 -exclude_-partial -copies -sc 0.98 -cds on the Y-chromosome and X-chromosome assemblies of both species using the X-chromosomes of the current assemblies of the Rambouillet ARS-UI_Ramb_v2.0[68] for sheep and the ARS_UCD_1.3[69] for cattle as reference. The liftoff result was filtered for only protein-coding genes (Supplementary Data S15 and S16). Homology-based gene annotation on the Y-chromosomes was done using protein sequences from cattle, sheep, mouse and the human T2T genome assemblies. Miniprot[70] was used with the --outc=0.7 option to output only hits with at least 70% coverage between the query and the target. The result was filtered with the alignment score (Supplementary Data S15 and S16) before manual curation with Apollo[71], a genome annotation plug-in for JBrowse[72].

### Repeat elements annotation

Repeats analysis was done on the Y-chromosomes with RepeatMasker version 4.1.2-pl[73] using a combined repeats library produced from the *Dfam* library (dfam.h5 version 3.7) and an older library (RepeatMasker.lib) in order to have a comprehensive database of repeat elements. Details of the command used are listed in Supplementary Methods 2.

### Comparison of the T2T Y-chromosomes with publicly available assemblies

We collected all the cattle and sheep genome assemblies where male samples were reported sequenced from NCBI. Out of a total of eleven assemblies published for cattle, seven assemblies came from male samples, but only Btaurus_INIA1 was not assembled to chromosome level. All the chromosome-level assemblies reported partial Y-chromosome assemblies except Bos_taurus_UMD_3.1.1 which had none (Supplementary Data S14). The current cattle reference genome

ARS_UCD1.3, does not have a Y-chromosome assembly since it is from a Hereford cow. We thus selected the Y chromosome from the Btau_5.0.1 (GCA_000003205.6) assembly being the longest at 43.3 Mb.

For the publicly available sheep assemblies, from 56 whole genome assemblies NWAFU_Friesian_1.0 and ASM2243283v1 were the two chromosome-level assemblies out of the 6 produced from male animals (Supplementary Data S15). Take note that assemblies ASM2243283v1, ASM2132593v1 and ASM2132590v on one hand, and assemblies ASM2270250v1 and ASM2270250v on the other hand were produced from the same biotypes and were thus not treated as unique assemblies. We selected the Hu Sheep assembly[14] which covered only the MSY of the sheep Y chromosome for comparison with our T2T Y-chromosome.

We aligned the T2T Y-chromosome assemblies to the publicly available assemblies with Mashmap[66] at a minimum identity of 90% and segment length of 5 kb.

### Defining the centromere
The centromeres of the Y-chromosomes were defined using satellite DNA repeats annotation Tandem Repeat Finder (TRF)[74], CENP-A enrichment analysis from CUT&RUN data, and epigenetic information of methylated Cytosine residues at the centromeric region.

### Satellite DNA annotation with Tandem Repeat Finder (TRF)
TRF was run on the two Y-chromosome assemblies according to Supplementary Methods 3.

The output of TRF was converted into a bed file and visually inspected on JBrowse to identify regions with tandem arrays of repeat elements. The 73 bp repeat sequence which was identified in this manner from the centromeric tandem array was aligned back to the cattle (Wagyu and Charolais) and sheep (Churro and Friesian) assemblies using Blastn[75] to estimate their relative abundance. The blast results were filtered with a minimum sequence coverage of 80% between the query and the target, and a minimum e-value of 1E-3.

### CENP-A enrichment analysis
CENP-A pull-down assay was produced from bovine satellite cells obtained from semeimbranosous muscle of a 53-day old commercial male Angus calf using the GeneTex antibody with catalog number GTX13939 and sequenced. Adapter sequences were trimmed from the raw reads using Cutadapt[76]. The trimmed high-quality reads were then aligned to the whole genome sequence of the Y chromosome-containing Wagyu cattle using Bowtie2[77] with default parameters asides "-k 100". The alignments were sorted with Samtools[65] and duplicates were removed with Picard toolkit (https://broadinstitute.github.io/picard/). Bedgraph files were created from the alignment files as input into the SEACR[78] peaks caller to call the CENP-A peaks. SEACR was run using the stringent and 10% filter options (Supplementary Methods 4).

### Methylated Cytosines analysis (CpG Methylation)
CpG methylation data was produced with ONT and PacBio HiFi reads using the manufacturer-prescribed protocols.

### Oxford Nanopore Technology (ONT). 
ONT-based methylated CpG sites were called in four main steps:
1. Methylated bases were called bases with Dorado using the R9_v0.3.4_v3.3_5hmc-5mc_A100mig model.
2. Fastq files were extracted from the *modbams* and the MM, ML tags were retained.
3. The fastq files were aligned to the reference assembly with Winnowmap[79]. The resulting sequence alignment map (.sam) file was filtered, converted to a bam file and indexed.
4. The bam file was parsed into modbam2bed to produce the bed file containing the methylated sites filtered using the 20% threshold for unmethylated bases and 80% for methylated bases.

**Pacific Biosciences (PacBio).** The prescribed PacBio pipeline for calling CpG methylated sites as stated in Supplementary Methods 5 was used in this study. The CpG methylation calls obtained from these steps were post-processed by first changing all the NANs in the methylation values to 0.00 s. The aggregate methylation levels for 5 kb and 10 kb bins were calculated using custom python scripts. The resulting data were plotted with an R script in RStudio to visualize the tracks of the methylation pattern. With visual inspection of the methylation tracks, the CDR was defined according to[49] as the region with a local dip in methylated Cytosines with respect to the surrounding sequence and flanking the active satellite array.

### Transcript level quantification on the Y-chromosomes
We obtained publicly available RNA-Seq data of transcripts quantification experiments from different tissues in order to measure the transcription activity of the genes on the Y-chromosomes.

Sheep: Two sets of transcriptome data were obtained for the transcription activity analysis on the sheep Y-chromosome.
i. PRJNA552574: RNA-Seq was used to investigate whether testis development and gene expression is altered by the exposure of sheep to gossypol in utero and during lactation[80]. We obtained RNA-Seq data from a total of 18 testes samples from 60-day old lambs with 9 in control and 9 in treated groups were used.
ii. PRJNA437085: This study used RNA-Seq to investigate the effects of dietary energy levels on Hu sheep testicular development[81]. RNA-Seq data from three replicates each of the two energy levels were obtained for our analysis.

Cattle: Two sets of transcriptome data were also used for the cattle.
i. PRJNA565682: Six samples were retrieved from the project in an experiment to profile the transcriptome of mature and immature testes[82]. Three replicates each of the mature (24-month old) and immature (1-day old) testes of Chinese Red Steppes cattle were used (Supplementary Data S8).
ii. The second set was RNA-Seq data obtained from the Lung, Spleen, Muscle and Liver tissues of the Wagyu_x_Charolais F1 offspring from which the cattle Y-chromosome was obtained.

Quality control steps were carried out on the raw.*fastq* files to remove adapter, poly-X and poly-G sequences using fastp[83] with the parameters -f 4 --trim_poly_g --trim_poly_x

STAR splice-aware aligner[84] was then used to align the reads from the multi-tissue RNA-Seq data to the whole genome assemblies of the Wagyu and the Churro from where the Y-chromosomes were obtained. The details of the commands used for this analysis are stated in Supplementary Methods 6. The output of this run is the number of reads aligned to each gene.

### Phylogenetic analysis
The sequence for each member of the ampliconic gene families were extracted from the coordinates using *bedtools getfasta* module. Multiple sequence alignment was performed using Mafft[85]. The output was imported to Unipro Ugene using the PhyML maximum likelihood tree building method with branch length optimized.

### Dot plots
The self-alignment dotplots of the Y-chromosome assemblies were produced for visualization with Moddotplot (https://github.com/marbl/ModDotPlot) using the program parameters *--no-bed --identity 85 -s 52*.

### Reporting summary
Further information on research design is available in the Nature Portfolio Reporting Summary linked to this article.

## Data availability

Source data are provided witht this paper. The T2T Y-chromosomes have been added to the current reference assemblies of cattle and sheep on NCBI. They can also be accessed independently in the NCBI under the accessions GCA_030378505.1 [https://www.ncbi.nlm.nih.gov/datasets/genome/GCA_030378505.1] for the cattle, and GCA_030512445.1 [https://www.ncbi.nlm.nih.gov/datasets/genome/GCA_030512445.1] for the sheep. The methylated cytosines data generated in this study have been deposited in the GEO database under the accession GSE262831 for sheep, and GSE263098 for cattle. The cattle CENP-A CUT&RUN data generated in this study has been deposited in the GEO database under the accession GSE262830. The RNA-Seq data used for the transcriptional activity analysis in this study are available in NCBI under accession code PRJNA565682 for cattle and PRJNA552574 and PRJNA437085 for sheep. Source data are provided with this paper.

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

## Acknowledgements

This research was supported by the intramural research program of the U.S. Department of Agriculture, National Institute of Food and Agriculture, Grant Number 2023-67015-39000, award number USDA-NIFA-2021-67016-33416, and Hatch grant no. IDA01566. T.P.L.S. was supported by the appropriated project 3040-31000-104-000-D, "Genomes to Phenomes in Beef Cattle Research" and B.D.R. was supported by the appropriated project 8042-31000-112-000-D, "Accelerating Genetic

Improvement of Ruminants Through Enhanced Genome Assembly, Annotation, and Selection" of the USDA Agricultural Research Service. S.K. and A.R. were supported, in part, by the Intramural Research Program of the National Human Genome Research Institute (NHGRI), National Institutes of Health (NIH). The results reported here were made possible with resources provided by the USDA shared compute clusters Ceres and Atlas as part of the ARS SCINet initiative. Any mention of trade names or commercial products is solely for the purpose of providing specific information and does not imply recommendation or endorsement by the U.S. Department of Agriculture. The USDA is an equal opportunity provider and employer.

## Author contributions

T.P.L.S., B.M.M., B.D.R., and T.A.O. conceived the project; B.D.R., S.K. and A.R. generated the assemblies; T.A.O., B.D.R. and M.R.S. contributed to the data analysis; G.M.B., K.M.D., M.R.S., T.S.H., A.L.S., J.W.T., P.V., N.E.C. and H.L.N. contributed to samples collection; K.L.K. and K.A.S. generated laboratory data; C.G.E. provided resources for data annotation; T.A.O. drafted, T.P.L.S., B.M.M. and B.D.R. revised the manuscript. All the authors read and approved the final version of the manuscript.

## Competing interests

T.A.O. and S.K. have received travel funds to speak at events hosted by Oxford Nanopore Technologies. The remaining authors declare no competing interests.
