## [Peer Review File · Nature Communications]

Telomere-to-telomere assemblies of Cattle and Sheep
Y-chromosomes uncover divergent structure and gene
contentREVIEWER COMMENTS

Reviewer #1 (Remarks to the Author):

In this study, the authors produced single contiguous high-quality Y-chromosome assemblies of cattle and sheep with the ONT ultra-long and PacBio Hifi reads, and subsequently analyzed the structure and organization of the genomic structures contained in these Y-chromosomes. In general, the manuscript is well written and in good shape. The generated resource and findings will be valuable for research in sex chromosome biology. The only drawback is that it mainly describes differences, but it is difficult to reflect the principles of Y chromosome evolution. I have no major concerns on the manuscript, except that some minor comments for the authors to consider in revision. Below are my comments and suggestions, which I hope will be helpful.

- 1、 The authors provided novel insights into the content and organization of the centromeric region of the chromosomes, but lacked characterization of the telomeric regions of these two Y-chromosomes.
- 2、 Adjust the order in which the attached Supplemental Figs and tables, according to the order in which they appear in the main text.
- 3、 Line 35, change "chromosome" to "chromosomes".
- 4、 Line 93, Supplementary Table S1 did not provide the contig information. "NG50" should be "N50".
- 5、 line273, change "higher-order repeat (HOR)" to HOR.
- 6、 Figure 1c. The pattern diagram of the Y chromosome appears to depict it as a mesocentric chromosome, which could lead to misunderstandings.
- 7、 Figure 2a. Please add quantity labels.
- 8、 Figure 3ab. Please use triangles of the same size to display repeat units in cattle and goat.
- 9、 Figure S4. The blue area representing the transposable elements BOV-A2 and BovB is not obvious.
- 10、 Figure S7a and Figure S8a The coordinate labels are blurry, making it difficult to read. Request a higher resolution image.
- 11、 Line385 change "the three previously unreported genes" to "the three previously unreported genes (LOC112445918, PRB1 and PRB3 ?)".
- 12、 Line 33 mentioned "18MYA", while in line 402 is "19MYA". What is the different between these two time?

Reviewer #2 (Remarks to the Author):

This paper presents the first T2T assembly of the Y chromosome of two ruminant species: *Bos Taurus* and *Ovis Aries*. This assembly shows a different evolution of the Y chromosome of these two species that diverged 18 MYA ago, suggesting different evolutionary pressures on the Y chromosome of these two species.

The paper is very complete and well discussed.

My main comment is that it is frustrating to compare the genomes of only two species. In this context, highlighting different evolutions does not make it possible to determine whether a particular pressure has been exerted on the Y chromosome of one or other ruminant species, or whether the Y chromosome of many ruminant species has recently undergone very different evolutions.

While a reference genome is available for many ruminant species, a study of the type presented in this paper requires as complete an assembly as possible, from telomere to telomere, including repeated zones of the genome, and this type of study is not yet available for many species. However, it seems that many of the authors of this paper are participating in another paper, available on Researchgate, which mentions at least partial assembly of the Y chromosome of *Ovis Canadensis*. It would be interesting to see at the very least whether a sufficient assembly of the Y chromosome of this species is available and to see whether it adds anything to the comparison presented in this paper between the Y chromosomes of *Bos Taurus* and *Ovis Aries*.

Alternatively, there may be data available for other species, even if not assembled telomere to telomere, that would suggest some clues to the evolution of the ruminant Y chromosome.

In data availability, you indicate the reference of Y chromosomes. You should also submit the X chromosome haplotypes of the two sequenced male individuals. This will provide a resource for pangenome analyses.

I did not find in the supplementary tables a table giving information on the sequencing depth by Illumina, Pacific Biosciences and Oxford Nanopore technologies used for these assemblies, and whether you used Oxford Nanopore ultra long read sequences. A few lines should be added in the results, possibly a and the discussion on this point.

Miscellaneous points

- L104 for sheep genome the size should be rounded as 19.3 and 17.8 kb, for cattle genome rounded as 14.7 and 20.3 (correct for the last one)
- L111: the ratio between the centromeres is 20 (precisely 20.7) and not 15
- L530: It is table S14 not S15
- L537: It is table S15 not S14

I have also suggestions or a better understanding of your results

Figure 1:

1. It could be organized more clearly. When we look at fig 1A we see 2 tracks in green and red, and just on the right, we see SINEs and LINEs, whereas the appropriate legend is far below, indicating genes and pseudogenes.
2. I would suggest you add on this Fig 1A a track with LINEs and possibly an other with SINEs (as on the paper from Rhie et al, on Y human chromosome).
3. The color used for centromere, should be more different than the one used for PAR and ampliconic sequences. The triangle below the pseudogene track, showing the centromere, is not presented in the legend.
4. On 1C, “HOR of a 73 bp monomer” and “composite repeat including a LINE and a SINE” is not at the same level, as in cattle the block at the same level is of 3.7 kb (including a monomer of 73 bp). This is a bit confusing.
5. I wonder, if 1C could be deleted. This is better presented in Fig 3, and to understand 1C, you should first look at fig 3.

Figure 2:

1. On fig 2A and 2B, use the same scales for genes and for pseudogenes, the differences will be clearer.
2. On 2B, the colors used to distinguish the ampliconic gene families is not enough different. Very difficult to distinguish TSPY, ZNF280 and RBMY
3. On 2B, it seems that the scale for protein-coding genes is not correctly located: 0 on the scale is at a higher level than the 0 on the drawing, and similarly the 8 on the scale is above the 8 on the drawing.
4. It seems that the gene density appears in integer values in cattle, but on sheep genome, it seems more variable with perhaps 6 levels between 0 and 3 for pseudogene (see for instance around 12 Mb). Is that correct ?
5. I feel that the representation for ampliconic gene family is very difficult to read. It is really much more understandable on fig 1 and fig 2 of Rhie et al paper on human chr Y.

Figure 3:

1. The indication “the segment are not to scale” follow a *, but, there is no * on the figure
2. Even if the drawing is not to scale, a less confusing scale could be used for cattle, as red and pink modules are drawn at the same scale, whereas spacer1 & spacer2 are 3-4 times larger than the 5mer & 6 mer modules.
3. It seems that Spacer1 & Spacer2 contains 4mers, so the rectangles shown in spacer 1 and 2 contains most of their sequences, whereas presently it represents less than 50%. Even not at scale, a more understandable drawing can be proposed
If you succeed to improve this drawing, use it also on fig 1C (if you decide to keep it).
4. It is also confusing to have red and pink boxes at the levels of HOR Unit, and of 73 bp monomer, whereas red and pink do not represent the same at both levels. It would be more understandable if you use a different family of colors for each level.
5. On Fig 3B, the size of centromere indicated is 118 kb, whereas it is 120.7 on table 1.

Table S2:

1. Non repetitive is in the list of repeat_element.
2. The sum of all element is 98.9 % on cattle and 102.9 % on sheep.
3. The sum of bp, 58.85 and 26.66 Mb is different from the total size indicated in table 1 (59.48 and 25.92 respectively), this difference is probably linked to the 98.9 and 102.9 % indicated above.

Fig S1:

1. Legend: there is no red box for cattle assembly, but a green box.

Reviewer #1 (Remarks to the Author):

In this study, the authors produced single contiguous high-quality Y-chromosome assemblies of cattle and sheep with the ONT ultra-long and PacBio Hifi reads, and subsequently analyzed the structure and organization of the genomic structures contained in these Y-chromosomes. In general, the manuscript is well written and in good shape. The generated resource and findings will be valuable for research in sex chromosome biology. The only drawback is that it mainly describes differences, but it is difficult to reflect the principles of Y chromosome evolution. I have no major concerns on the manuscript, except that some minor comments for the authors to consider in revision. Below are my comments and suggestions, which I hope will be helpful.

1、 The authors provided novel insights into the content and organization of the centromeric region of the chromosomes but lacked characterization of the telomeric regions of these two Y-chromosomes.

Information on the characterization of the telomeric regions has been included in the main text between Line 110 and Line 115.

“The telomeric regions were characterized by a tandem repeat of the 6-mer sequence CCCTAA on the p-arm and the reverse complement TTAGGG on the q-arm of both Y-chromosomes. The cattle Y-chromosome contained 2,447 copies of the complete sequence on the p-arm and 3,387 copies on the q-arm. Similarly, the sheep Y-chromosome harbored 3,211 copies of the 6-mer on the p-arm while the q-arm contained 2,962 copies.”

2、 *Adjust the order in which the attached Supplemental Figs and tables, according to the order in which they appear in the main text.*

The supplemental figures and tables have been arranged in their order of appearance in the main text.

3、 *Line 35, change "chromosome" to "chromosomes".*

This has been changed, see Line 35.

“These Y chromosomes have been added to the current reference assemblies in GenBank opening new opportunities for the study of evolution and variation while supporting efforts to improve sustainability in these important livestock species that generally use sire-driven genetic improvement strategies.”

4、 *Line 93, Supplementary Table S1 did not provide the contig information. “NG50” should be “N50”.*

The assembly statistics have been provided in Supplementary Table S2 while NG50 has been corrected to N50. See Lines 96-98

“The Y-containing paternal haplotype assemblies were highly contiguous with contig N50 of 96.68Mb (cattle) and 108.17Mb (sheep) (Supplementary Table S2)”

5、 *line273, change "higher-order repeat (HOR)" to HOR.*

This has been changed, see Line 283.

“The 73bp monomeric satellite unit was arranged into a HOR structure wherein a total of 364 copies of the 3.7kb unit spans the 2.5Mb centromeric region (Figure 3A)”

6、 *Figure 1c. The pattern diagram of the Y chromosome appears to depict it as a mesocentric chromosome, which could lead to misunderstandings.*

The image has been modified to reflect the true chromosome type based on the position of the centromere.

7、 *Figure 2a. Please add quantity labels.*

The quantity labels have been added to the plot.

8、 *Figure 3ab. Please use triangles of the same size to display repeat units in cattle and goat.*

The sizes of the triangles have been adjusted to be uniform for both cattle and sheep.

9、 *Figure S4. The blue area representing the transposable elements BOV-A2 and BovB is not obvious.*

This region representing the transposable elements appear difficult to distinguish because it is very few and each unit is very small relative to the other regions. The image resolution has however been improved and a good pair of contrasting colors used in order to better highlight the region representing the transposable elements.

Please note that with the rearrangement of the supplementary figures, this figure is now S9.

10、 *Figure S7a and Figure S8a The coordinate labels are blurry, making it difficult to read. Request a higher resolution image.*

The resolution of the images has been improved to clearly show the labels.

11、 *Line385 change "the three previously unreported genes" to "the three previously unreported genes (LOC112445918, PRB1 and PRB3 ?)".*

This has been changed, see Line 395.

“On the sheep Y-chromosome however, the three previously unreported genes (*LOC112445918*, *PRB1* and *PRB3*) did not have homologs on the X-chromosome PAR, nor on the autosomes”

12、 *Line 33 mentioned "18MYA", while in line 402 is "19MYA". What is the different between these two time?*

The 18MYA was an error and has been corrected according to the information contained in reference 48 on line 412.

Reviewer #2 (Remarks to the Author):

This paper presents the first T2T assembly of the Y chromosome of two ruminant species: Bos

Taurus and Ovis Aries. This assembly shows a different evolution of the Y chromosome of these two species that diverged 18 MYA ago, suggesting different evolutionary pressures on the Y chromosome of these two species.

The paper is very complete and well discussed.

My main comment is that it is frustrating to compare the genomes of only two species. In this context, highlighting different evolutions does not make it possible to determine whether a particular pressure has been exerted on the Y chromosome of one or other ruminant species, or whether the Y chromosome of many ruminant species has recently undergone very different evolutions.

While a reference genome is available for many ruminant species, a study of the type presented in this paper requires as complete an assembly as possible, from telomere to telomere, including repeated zones of the genome, and this type of study is not yet available for many species. However, it seems that many of the authors of this paper are participating in another paper, available on Researchgate, which mentions at least partial assembly of the Y chromosome of Ovis Canadensis. It would be interesting to see at the very least whether a sufficient assembly of the Y chromosome of this species is available and to see whether it adds anything to the comparison presented in this paper between the Y chromosomes of Bos Taurus and Ovis Aries.

Alternatively, there may be data available for other species, even if not assembled telomere to telomere, that would suggest some clues to the evolution of the ruminant Y chromosome.

We agree with you that a truly comprehensive view of the evolution of the Y-chromosome will require more than the comparison of two species. Indeed, the Ruminant T2T (RT2T) Consortium plans to assemble the genomes of dozens of species within this clade and perform large-scale comparative analyses across all autosomes and sex chromosomes. This ambitious project is expected to take another year or two before genome sequencing is completed. Because cattle and sheep are two of the most important agricultural livestock species for genomic selection, they were our first targets in the project. The reference genomes of cattle and sheep lacked Y-chromosomes, and the consortium agreed in consultation with NCBI that getting complete Y chromosomes added to the references would be an important immediate contribution to ongoing global research and breeding efforts.

The results presented in this study are hypotheses that were generated from the observations in this comparison and can be tested when we have the full set of species from the ongoing efforts of the RT2T consortium. In our opinion, adding a single species such as Bighorn which is closely related to domestic sheep to this current study would not provide any substantive test of the hypotheses that have been raised in the current manuscript.

Repeat sequence provides insights into the evolutionary trajectory of sex chromosomes, and by extension, a species. Indeed, the hypothesis we present in this manuscript depended entirely on the highly repetitive regions of the chromosomes, which are lacking in non-T2T assemblies

present in GenBank. Even the Bighorn Y chromosome presented in our pre-print is not T2T. Therefore, using those assemblies would not provide additional clarity or allow us to validate our hypotheses. To further highlight the points made on lines L417-L419, L437-L439, and L443-L445, we have added the following statement to the conclusion on L445-L446 “Our hypotheses can be tested when the full suite of ruminant Y-chromosomes is available from the RT2T consortium.”

In data availability, you indicate the reference of Y chromosomes. You should also submit the X chromosome haplotypes of the two sequenced male individuals. This will provide a resource for pangenome analyses.

We agree that X-chromosome assemblies are important. The Y chromosomes were promptly added to the current reference genomes of cattle and sheep in order to make these sex-determining chromosome available to the research community since the current reference genomes lack Y-chromosomes. However, as we stated in the manuscript (see L85), these chromosomes were obtained from draft assemblies that are still undergoing manual curation and polishing. We intend to make the complete assemblies available on NCBI as soon as curation and polishing are completed.

I did not find in the supplementary tables a table giving information on the sequencing depth by Illumina, Pacific Biosciences and Oxford Nanopore technologies used for these assemblies, and whether you used Oxford Nanopore ultra long read sequences. A few lines should be added in the results, possibly a and the discussion on this point.

A table has been provided in the supplementary tables (Supplementary Table S1) highlighting this information for the mix of technologies used for the sequencing. This information has also been included in the results section of the manuscript (L92-L96).

“For cattle, about 626.4 Gbp raw ONT reads with 129.4 Gbp of UL reads at 101x and 21x respective reads depth were used for the genome assembly, while 328.6 Gbp of PacBio HiFi raw reads at 53x were used (Supplementary Table S1). Similarly for sheep, 600.8 Gbp of raw ONT reads at 106x with 15x of 84.6 Gbp UL reads were combined with 258.1 Gbp at 46x for the assembly (Supplementary Table S1).”

Miscellaneous points

- *L104 for sheep genome the size should be rounded as 19.3 and 17.8 kb, for cattle genome rounded as 14.7 and 20.3 (correct for the last one)*

These have been corrected, see L107-L110).

“Telomere sequences were located at the distal ends of the two chromosomes – 14.7kb and 20.3kb from the p- and q- arms of the cattle and 19.3kb and 17.8kb from the p- and q- arms of the sheep, (Table 1, Figure 1A) indicating the completeness of the single-contig assemblies.”

- *L111: the ratio between the centromeres is 20 (precisely 20.7) and not 15*

This has been corrected, see L120.

“The 120.76kb sheep Y-chromosome centromere at 8.03Mb from the end of the short arm is 20 times smaller than the 2.52Mb-long centromere on the cattle located at 14.12Mb from the end of the shorter arm of the chromosome (Table 1, Figure 1A)”

- *L530: It is table S14 not S15*

This has been corrected, see L541.

“All the chromosome level assemblies reported partial Y-chromosome assemblies except *Bos_taurus_UMD_3.1.1* which had none (Supplementary Table S17)”.

Please note that with the addition of a table at the start and the rearrangement of the supplementary tables it is now Table S17.

- *L537: It is table S15 not S14*

This has been corrected, see L548.

“For the publicly available sheep assemblies, from 56 whole genome assemblies *NWAFU_Friesian_1.0* and *ASM2243283v1* were the two chromosome-level assemblies out of the 6 produced from male animals (Supplementary Table S18)”.

Please note that with the addition of a table at the start and the rearrangement of the supplementary tables it is now Table S18.

Figure 1:

1. *It could be organized more clearly. When we look at fig 1A we see 2 tracks in green and red, and just on the right, we see SINEs and LINEs, whereas the appropriate legend is far below, indicating genes and pseudogenes.*

We apologize for the confusion that this representation may have created. The SINEs and LINEs you referred to on the right belongs to the legend of fig 1B showing the repeats element composition of the two chromosomes. See the response in number 2 below.

2. *I would suggest you add on this Fig 1A a track with LINEs and possibly an other with SINEs (as on the paper from Rhie et al, on Y human chromosome).*

The tracks for the LINEs and the SINEs have been added to figure 1.

3. *The color used for centromere, should be more different than the one used for PAR and*

ampliconic sequences. The triangle below the pseudogene track, showing the centromere, is not presented in the legend.

The color for the centromere has been changed to red which contrasts the other colors on the tracks. The triangle was added to further highlight the location of the centromere especially for the Sheep-Y which is very small. A triangle has now been added to the legend for centromere annotation.

4. On 1C, "HOR of a 73 bp monomer" and "composite repeat including a LINE and a SINE" is not at the same level, as in cattle the block at the same level is of 3.7 kb (including a monomer of 73 bp). This is a bit confusing.

Thank you for the comment, we have adjusted the block sizes on Figure 1C.

5. I wonder, if 1C could be deleted. This is better presented in Fig 3, and to understand 1C, you should first look at fig 3.

Thank you for the suggestion. Figure 1C shows a high-level organization of the cattle and sheep centromeric repeat as a HOR for cattle and a composite repeat for sheep. Although the details of the content of the centromeres are presented in Figure 3, we have used Figure 1C in presentations to different audiences and realized that displaying this high-level image of the structure of the centromeres prior to the detailed organization provided a better understanding of the information.

Figure 2:

1. On fig 2A and 2B, use the same scales for genes and for pseudogenes, the differences will be clearer.

The scales have been adjusted accordingly.

2. On 2B, the colors used to distinguish the ampliconic gene families is not enough different. Very difficult to distinguish TSPY, ZNF280 and RBMY

A new set of contrasting colors has been applied to these ampliconic genes families to better distinguish them in the image.

3. On 2B, it seems that the scale for protein-coding genes is not correctly located: 0 on the scale is at a higher level than the 0 on the drawing, and similarly the 8 on the scale is above the 8 on the drawing.

The position of the scale has been properly aligned to the level of the drawing.

4. It seems that the gene density appears in integer values in cattle, but on sheep genome, it seems more variable with perhaps 6 levels between 0 and 3 for pseudogene (see for instance around 12 Mb). Is that correct ?

In fact, the gene density across the chromosome is comprised of the gene count in each window as integer values for both cattle and sheep. The appearance of continuity in the figure is a function of the scale, the window size, and the number of windows used for the calculation. The figure legend has been adjusted to better reflect this. “(B) The gene density showing the number of ampliconic genes across the Y-chromosomes calculated in 100kb bin sizes highlighting the loci of the protein-coding genes on the upper tracks and the pseudogenes on the lower tracks.”

5. I feel that the representation for ampliconic gene family is very difficult to read. It is really much more understandable on fig 1 and fig 2 of Rhie et al paper on human chr Y.

In Rhie et al., a single Y chromosome was presented for human where each ampliconic gene (only those which formed composite repeat arrays) was represented on separate tracks to highlight their loci. In our comparison of two Y-chromosomes, we did not have the latitude to present the ampliconic genes on separate tracks. This is why the detailed representation of these genes on separate tracks was provided in Supplementary Figure S4. We think that the comparative analysis that we show with the figure is more pertinent to the hypotheses that we propose.

Figure 3:

*1. The indication “the segment are not to scale” follow a *, but, there is no * on the figure.*

The inclusion of the asterisk was initially meant to be applied to fig 3B with the conspicuously disproportionate segments. Since the segments in the two figures are not to scale, the asterisk has been removed such that the statement applies to both figures now.

2. Even if the drawing is not to scale, a less confusing scale could be used for cattle, as red and pink modules are drawn at the same scale, whereas spacer1 & spacer2 are 3-4 times larger than the 5mer & 6 mer modules.

The scale of the spacers and segments have been adjusted to better reflect the relative differences between them.

3. It seems that Spacer1 & Spacer2 contains 4mers, so the rectangles shown in spacer 1 and 2 contains most of their sequences, whereas presently it represents less than 50%. Even not at scale, a more understandable drawing can be proposed

The image has been improved to convey the relative sizes of the segments.

If you succeed to improve this drawing, use it also on fig 1C (if you decide to keep it).

The relative sizes of the segments on Figure 1C have been modified to reflect the relative sizes of the segments.

4. It is also confusing to have red and pink boxes at the levels of HOR Unit, and of 73 bp

monomer, whereas red and pink do not represent the same at both levels. It would be more understandable if you use a different family of colors for each level.

Thank you for this observation. The closeness of the colors was to highlight that the segments share some degree of sequence similarity, hence the use of a lighter shade of red (rendered as pink) for the spacer segments with lower sequence similarity to the red 5-mer and 6-mer segments. We have contrasted the colors with red and blue for the segments. We also included a statement in the figure legend as follows: “The arrows representing the monomers have different color densities to highlight the divergence between them.”

5. *On Fig 3B, the size of centromere indicated is 118 kb, whereas it is 120.7 on table 1.*

The correct length of 120.7 kb has been entered on the figure.

Table S2:

1. *Non repetitive is in the list of repeat_element.*

This was supposed to be labeled as the total of all repetitive elements and not “Non_repetitive”. The proper label has been inserted.

2. *The sum of all element is 98.9 % on cattle and 102.9 % on sheep.*

The last record which was labeled “Non_repetitive” was supposed to be the total of all repetitive elements and this has been corrected.

3. *The sum of bp, 58.85 and 26.66 Mb is different from the total size indicated in table 1 (59.48 and 25.92 respectively), this difference is probably linked to the 98.9 and 102.9 % indicated above.*

Yes, the difference is linked to the percentage difference above as the last entry was supposed to be the total of all the repetitive elements. The correct label has been applied.

Fig S1:

1. Legend: there is no red box for cattle assembly, but a green box.

This has been changed accordingly in the caption.

REVIEWERS' COMMENTS

Reviewer #1 (Remarks to the Author):

The authors have addressed all my concerns.

Reviewer #2 (Remarks to the Author):

Thank you for this new version, in which you have taken on board the suggestions I made. I have the feeling that the figures will be easier for everyone to understand, and will make it easier to understand your results.

I agree on the difficulty of including at least a 3rd species, knowing that the completeness of the sequence of this 3rd species would be inferior for the moment.

I still have a question about the legend to Figure 1A. Is the density of SINES and LINES really calculated over a 1 kb window? How is it possible to have more than 10 LINES per kb?

Reviewer #2 (Remarks to the Author):

Thank you for this new version, in which you have taken on board the suggestions I made. I have the feeling that the figures will be easier for everyone to understand, and will make it easier to understand your results.

I agree on the difficulty of including at least a 3rd species, knowing that the completeness of the sequence of this 3rd species would be inferior for the moment.

“I still have a question about the legend to Figure 1A. Is the density of SINES and LINES really calculated over a 1 kb window? How is it possible to have more than 10 LINES per kb?”

Thank you for your feedback.

The bin size used is indeed 10kb (`bin.size=1e4`) as can be seen in the following CMPlot R code snippet used to generate the image.

```
CMplot(dataCM,plot.type="d",bin.size=1e4,chr.den.col=c("darkgreen", "yellow",  
"red"),file="jpg",file.name="cattle_LINES",dpi=300,  
main="LINES and SINES",file.output=TRUE,verbose=TRUE,width=9,height=3)
```

The initially stated 1kb was an error and has been corrected on the legend to reflect the correct bin size of 10kb.